# Modeling Trajectories Obtained from External Sensors for Location Prediction via NLP Approaches

**DOI:** 10.3390/s22197475

**Published:** 2022-10-02

**Authors:** Lívia Almada Cruz, Ticiana Linhares Coelho da Silva, Régis Pires Magalhães, Wilken Charles Dantas Melo, Matheus Cordeiro, José Antonio Fernandes de Macedo, Karine Zeitouni

**Affiliations:** 1Insight Data Science Lab, Federal University of Ceará, 60440-900 Fortaleza, Brazil; 2Laboratoire DAVID, University of Versailles Saint-Quentin-en-Yvelines, 78035 Versailles, France

**Keywords:** trajectory modeling, representation learning, sensors trajectory, trajectory embedding, location prediction, trajectory prediction

## Abstract

Representation learning seeks to extract useful and low-dimensional attributes from complex and high-dimensional data. Natural language processing (NLP) was used to investigate the representation learning models to extract words’ feature vectors using their sequential order in the text via word embeddings and language models that maintain their semantic meaning. Inspired by NLP, in this paper, we tackle the representation learning problem for trajectories, using NLP methods to encode external sensors positioned in the road network and generate the features’ space to predict the next vehicle movement. We evaluate the vector representations of on-road sensors and trajectories using extrinsic and intrinsic strategies. Our results have shown the potential of natural language models to describe the space of features on trajectory applications as the next location prediction.

## 1. Introduction

The extensive use of geolocation devices allows for the collection of a large volume of data from the trajectories of moving objects. Many machine learning tasks can benefit from this kind of data, such as real-time mobility events’ monitoring (e.g., detecting typical traffic flows, predicting traffic jams, or predicting the next location in road networks).

The challenge of determining the next important location of a moving object based on previous trajectories is known as location prediction (or trajectory prediction). Location prediction has drawn the researchers’ interest in recent years due to its various practical applications, including traffic management, police patrol, and tourism recommendation. This paper tackles the problem of learning how trajectories are represented for location prediction in the context of external, fixed, and sparse on-road sensors (e.g., traffic surveillance cameras) to capture vehicles’ trajectories.

Representation learning is an essential concept in machine learning, transforming input data into a useful format using a learning algorithm. Representation learning for trajectory modeling is concerned with building statistical or machine learning models of the observed trajectories of vehicles or people. Such models may have different uses: computing the probability of observing a given trajectory for anomaly detection [1]; estimating the importance of different characteristics that drivers may consider relevant when following a trajectory; recovering sparse or incomplete trajectories, such as those observed from external sensors [2]; aiding drivers in choosing an optimal route from an origin to a destination; the online prediction of the next location of a vehicle given its current location, [3] which is the application studied in this paper.

We evaluate representation learning methods to generate the feature space and predict the next vehicle movement, which will allow for the displacement of a vehicle among the video surveillance cameras to be predicted. In contrast with most early research, we forecast the next movement of objects as external sensors on the road network record their positions. It is possible to determine the trajectories of the moving objects as a sequence of sensor positions, assuming that each sensor record contains sufficient information to uniquely identify the associated moving object, such as identifying vehicles by their license plates.

In Natural Language Processing (NLP), many works [4,5,6,7,8] have proposed representation learning models based on neural networks to extract the features of words, or sentences, from their sequential order of words, while maintaining their semantic meaning. We take inspiration from NLP word representation methods that model the semantics of a word and its similarity with other words, by observing the many contexts of the word’s use in the language. Geometric distances between word vectors reflect semantic similarities and difference vectors encode semantic and syntactic relations [9].

Thus, we aim to investigate the following research questions:RQ1: Could NLP embedding models, more specifically, language models and word embeddings, be used to represent the vector space of trajectories in location prediction tasks using recurrent architectures?RQ2: Are the representations of sensors/locations from representation models in NLP able to capture their context, i.e., the closest sensors/locations, both in terms of road distance and connectivity?RQ3: Could trajectory representations from NLP embedding models adequately capture trajectories’ similarities?

State-of-the-art papers include the adoption of neural embeddings for model locations, points of interest (PoIs) [10,11,12] or to learn the temporal interactions between users and items [13]. The papers [14,15,16] also focus on the next location prediction task from trajectories obtained by external sensors and assume that a road network constrains object movements. Their trajectories report predefined positions (i.e., sensors’ location) on the road network, which makes trajectories much more sparse than the usual GPS raw trajectories. Thus, the location prediction is different from (i) the papers that consider GPS trajectories, which occur in continuous locations [3,17], and (ii) the next stop location prediction, which is usually applied to points of interest or event places as [18,19,20,21], since the prediction involves movement rather than stops. In addition, none of the previous works investigated whether the embedding encoders leverage the location prediction models or how the embedding space represents the spatial relationships between the sensors and the trajectory’s similarities.

The rest of this article is structured as follows. Section 2 explains the main concepts needed to understand this paper. Section 3 introduces the related work. Section 4 discusses the data, the NLP methods for obtaining embedding vectors, and the next location prediction architecture we used to achieve our goals. Section 5 presents and discusses the experimental evaluation. Section 6 concludes this work and suggests future developments.

## 2. Preliminaries

The Public Security Secretariat and Social Defense (SSPDS, acronym in Brazilian Portuguese ) of the state of Ceará, in Brazil, has developed the Approach Police System (SPIA, acronym in Brazilian Portuguese) to deal with the mobility of criminals. SPIA uses the video surveillance system and teams of police actions, especially motorcycle patrol. SPIA processes the data obtained by cameras equipped with a license plate recognition (LPR) system in real-time. SPIA checks whether each captured vehicle plate is related to a declared theft. If so, it informs the Integrated Police Operation Center (CIOPS, acronym in Brazilian Portuguese), where a vehicular approach police action is executed. Next, CIOPS operators plann how to approach the stolen vehicle through a visual examination of the video-monitoring cameras scattered around the city. Implemented in 2017, SPIA has led to a 48% reduction in vehicle theft actions in the state of Ceará. Figure 1 depicts a map of Fortaleza and its metropole region, with the surveillance cameras (red points) used by SPIA in 2019 plotted on that map. Despite the efficiency of SPIA, one of the most complex activities is predicting the trajectory of vehicle movement. At present, this is carried out manually, with several police agents simultaneously checking the video monitoring cameras. The modus operandi also requires the availability of several police vehicles scattered throughout the city, which must be dedicated to the service in question. Consequently, this current mode of operation is costly and subject to errors.

This paper considers the context in which trajectories are obtained from external sensors placed at fixed positions on roadsides, such as the ones collected by the SPIA. The moving objects have a unique identifier, for example, a vehicle’s license plate. The external sensors on the road network register the passage of moving objects.

Sensors may fail to capture the license plate and then produce incomplete trajectories with missing observations [14,15]. Furthermore, sensors are spatially sparse and not equally distributed, producing sparse trajectories. Figure 2 presents the histogram of road distances from one sensor to the nearest sensor (Figure 2a) and the third nearest sensor (Figure 2b) of our dataset. On average, the sensors are 440 m apart from their nearest neighbor. The nearest sensor is at least 600 m away from more than 25% of the sensors. Analyzing the distances between the third closest sensors, more than 40% of all sensors have their top three closest sensors located more than one kilometer away.

We present some definitions to support our problem statement in the same way as in our previous work [14,16].

**Definition** **1**(Trajectory). *A trajectory is a function F(t), that returns the location of a moving entity (e) at a given time t. In other words, it is a finite set of chronologically sequenced points t1⟶t2⟶⋯⟶tn. F(t)=(x,y), where (x,y) is the spatial information, latitude and longitude, where the object was at time t.*

**Definition** **2**(External Sensor Observation). *When the sensor registers a vehicle’s passage, it produces the tuple o=(m,s,t) where m is the identifier of the moving object, s is the sensor identifier, and t is a timestamp.*

**Definition** **3**(External Sensor Trajectory). *Let O be the set of observations generated by a set of sensors S. Let $O_{m}\subset O$ be the set of observations related to the moving object m. We define an External Sensor Trajectory (EST) of a moving object as the sequence of observations $o_{1}=\left(m,\;s_{1},\;t_{1}\right)⟶o_{2}=\left(m,\;s_{2},\;t_{2}\right)⟶\cdots ⟶o_{j}=\left(m,\;s_{j},\;t_{j}\right)$, such that $\forall i,1\leq i\leq j,o_{i}\in O_{m}$ and $t_{i}<t_{i+1}$.*

Now, we can define the problem that this paper tackles.

**Definition** **4**(EST Prediction). *Let G be the street network, S be the set of external sensors implanted on G, O be the set of historical observations produced by S, and $T_{EST}$ be the set of historical trajectories derived from O. Given the latest n observations of a driver m, $o_{i}=\left(m,\;s_{i},\;t_{i}\right)⟶o_{i+1}=\left(m,\;s_{i+1},\;t_{i+1}\right)⟶\cdots ⟶o_{i+n-1}=\left(m,\;s_{i+n-1},\;t_{i+n-1}\right)$, the problem consists of predicting the next sensor, $s_{i+n}$, that is to be visited by m.*

In general terms, the problem is predicting the next sensor from a given partial trajectory. The research questions presented in the last section that guides this work focus on investigating the natural language models to represent the trajectory sensors and exploring whether the prediction model improves.

We can observe a substantial similarity between natural language and EST. First, natural language and EST can be approximated by context. Given the context, we can predict the next word in the natural language.

Likewise, a road network constraints the movement of drivers, so their observation only occurs at fixed (predefined) positions (i.e., sensors’ location) on the road network. Thus, given a sequence of external sensors crossed by a driver, we can predict the next sensor on the driver’s trajectory.

Both domains can be viewed as time-dependent series. Regardless of the language, the dictionary can provide multiple choices that could succeed a word in a sentence. Likewise, there are various choices of paths in the driver’s trajectory; therefore, there are multiple possibilities for external sensors’ observation.

To motivate the use of natural language models in the representation learning of trajectory sensors, we verify that the frequency of trajectory sensors approximately follows a Zipf Law. The Zipf Law controls the word frequency distribution in natural language [22]. In [10], the authors observed the same Zipf’s Law behavior on human mobility habits and used this analogy to apply natural language models to learning representation for living habits. Figure 3 (observed in the dataset utilized in our experiments) shows the distribution of sensor observations, which roughly follow Zipf’s Law, as expected. In the plot, axis *y* is the frequency with which each sensor appears in trajectories; axis *x* is the frequency rank, where rank *k* is the kth most frequent sensor.

## 3. Related Works

In what follows, we split the related works into two categories: representation learning, i.e., models that learn how to extract useful information from data that help to build classifiers or other predictors. We focus on works that use NLP applications of representation, learning to deal with trajectory data. The second category is papers that tackle the problem of location prediction.

### 3.1. Representation Learning for Mobility Data

Representation learning refers to the machine learning methods that seek to transform complex, high-dimensional (and often redundant) representations into an effective and low-dimensional representation while preserving the information embedded in the raw data [23].

Several works have proposed representation learning for different applications as a modeling of the sequential interactions between users and items/products for social networks or e-commerce, where each user/item can be embedded in a Euclidean space, and its evolution can be modeled with an embedding trajectory in this space. JODIE [13] learns the temporal iterations between users and items via embeddings, through a coupled recurrent neural network, where the item embeddings are based on learning a future user embedding projection.

Habit2vec [10] models a person’s habit as a vector that upgrades the word2vec model according to the particular characteristics of trajectories. The paper aims to find the similarities between living patterns that engage in similar behavior at similar times instead of staying in geographically neighboring locations. Habit2vec trains the trajectories into a three-layer neural network model, Continuous Bag-of-Words (CBOW), to learn the embedding vectors of habits. After modeling the living habit into an embedding in a single space, the authors use classical clustering methods, such as K-Means and density-based methods, to cluster similar living habits.

In [24], the author proposes a time-aware trajectory embedding model in next-location recommendation systems to deal with sequential information and data sparsity problems. The user’s preference should reflect the evolving characteristics of user interests. The work [25] presents a sequence-to-sequence model to learning trajectory representations and compute the similarities between trajectories. The approach generates low sampling trajectories from high sampling trajectories without changing the underlying route. During training, the encoder embeds the low sampling trajectory in a vector representation, and the decoder tries to recover the original trajectory using the embedded vector. To incorporate spatial proximity in the model, a spatial proximity loss function was proposed that penalizes the error as much as the distance between spatial cells.

Mob2Vec [12] is a framework for learning representations of human mobility using trajectories obtained from Call Details Records (CDR). Mob2Vec uses Sqn2Vec [26], an unsupervised approach based on Paragraph Vector (PV) [27]. Mob2Vec firstly summarizes trajectories, removing irrelevant and noisy locations. Then, Mob2Vec obtains the vector representations of trajectories using Sqn2Vec. Finally, this aggregates the vectors of trajectories from the same user to obtain a unique representation.

TraceBERT [11] addresses the problem of location-based trajectory modeling. TraceBERT infers the lack of spatial observations of moving objects based on the previously visited locations along the trajectory. The proposed solution models each location as a word in the Bidirectional Encoder Representations from Transformers (BERT) model [28]. The BERT model works by masking certain words over the text, and tries to predict them based on the context provided by the unmasked words. As in this work, TraceBERT trains the BERT model on trajectories rather than text, masking the locations along the path and using the BERT model to infer the missing locations. Unlike TraceBERT, this work applies the NLP models to obtain the pre-trained embedding of sensors in order to incorporate them into a location prediction model. We evaluate how the embedding representation impacts the performance of predictors, both with and without fine-tuning. Additionally, we investigate how the learned representation can capture the spatial and connectivity relationships between sensors and trajectory similarities.

### 3.2. Prediction Models for Location Prediction

Location prediction uses the historical tracking of users to learn mobility patterns and predict a user’s future location given its most recent previous partial trajectory. Several researchers have studied this problem due to its usability on many location-based services, such as traffic control, route planning, and recommendation, to name a few.

MyWay [29] predicts human movements based on a single user’s behavior, a global model based on all user behaviors, and a hybrid approach. Personal mobility profiles capture user routines, and is applied to define routines as the representative trajectories of each cluster. An individual profile is a set of one user’s routines, and a global profile is a set of all individual profiles. For the predictions, they match the current trajectory with the profiles. TPRED [17] predicts the next stop of a moving object using a prefix tree, in which the nodes represent the object’s permanence in a specific location within a temporal partition, and the edges represent the observations of movements between nodes.

SERM [19] is a model based on RNN that learns from check-in tweets data to predict the next location. SERM uses trajectory data (user, location, time) and semantic features (keywords). For the location feature, SERM performs the discretization of the geographic space of cities into a grid of cells. The temporal feature is divided into time slots, and the semantic information of the trajectory is recovered from the hashtags of tweet check-ins posted by users at the locations. DeepMove [3] is a location prediction framework in the next time window. It contains a recurrent module that seeks to capture complex dependencies in the most recent trajectory and a historical attention module to capture the regularity and periodicity of movement in long historical records. The paper [30] proposes a group-based approach to predict the next stop of a single user. The trajectories were obtained from the movements of a group of travelers in the indoor environment. Profile information (such as gender and age) is used to classify people into groups. The method mines sequential rules from significant sequential patterns and then estimates the probability of visiting a location based on the recent movement of the user and his group.

A model to predict user movement in the next few minutes or hours based on a long-term spatiotemporal memory (LSTM) is presented in [31]. The LSTM models the context of the visit by identifying areas of interest (AOI), which represent relevant areas of geographic space, such as commercial areas and shopping centers. The proposed architecture modifies the internal operations of the LSTM by adding spatiotemporal factors to the network gates. TTDM [32] explores a set of historical trajectories to build a weighted graph in which the locations are the nodes, and the transitions between locations are the edges. TTDM predicts the next location based on the shortest travel time between candidate locations and integrates the strategy based on travel time with the Markov Model presented by [33] through linear interpolation.

The paper [15] defines EST Prediction problem and shows an in-depth study of the challenges of analyzing trajectories derived from external sensors, such as the problem of incompleteness and data scarcity. The general goal of [15] is to predict the next sensor from the trajectories that pass by sensors located on the road network. The [15] approach is RNN-based. To deal with the problems of incompleteness and scarcity, different data imputation methods are used and compared. The paper [14] presents a multitask neural network model based on space and time to tackle the same problem addressed by [15]. Next-location prediction in an external sensor network is also addressed in [34], which introduces a deep learning model for the historical trajectories obtained by traffic surveillance devices, deployed along with a city’s street network. The model incorporates semantic features, such as climatic and traffic flow information.

Location prediction from external sensors has to deal with trajectory data that are more sparse than the usual GPS raw trajectories with continuous locations [3,17,29]. As in the next stop prediction [18,19,20,21], sensors’ locations are discrete. However, additional information (such as user profile, user id, or semantic tags of the POIs) on those used by [19,30] are not available for our task.

In contrast to the previous works, we examine how sensor representations leverage the EST prediction task for a recurrent network model. We also show how the representation derived for sensors and trajectories performs in terms of trajectory similarity, as well as the distance and connectivity of sensors in the road network.

## 4. Data and Methods

This section explains the methodology used in this work to investigate the use of representation learning architectures for NLP on external sensors’ trajectories.

Firstly, we represented the trajectories as sequences of sensor labels. Since we want to evaluate these methods’ potential to capture the spatial representation, we discarded the temporal feature. We labelled each sensor using a geocode hash function applied to its geolocation.

Then, we trained a text representation learning model to learn a representation of the labels’ sensors. Mapping these to the text representation, we considered one sensor label as one word, a trajectory as a text, and the set of trajectories as the corpus of documents. The output of this step is a set of embedding vectors, with each one representing a sensor label.

We hypothesized that the vector embedding captured a relationship between sensors, such as spatial proximity or frequent use together. We experimented with the embedding vectors using different embedding models from NLP, such as Word2Vec [4] and BERT [28].

### 4.1. Data Preparation

We used the same dataset and pre-processing steps as in our previous work [16]. The trajectory data came from a real traffic monitoring application, which had 489 sensors monitoring vehicles at Fortaleza city in Ceará, Brazil. The dataset contained 2990 stolen vehicles and 22,530 sensor observations collected from January 2019 to June 2019. Each record included the vehicle license plate, timestamp, latitude, longitude, and speed. We used the sensor’s location to infer its identification.

We segmented the trajectories in order to identify trips. Firstly, the trajectories were segmented by day, where the points from the same vehicle on the same day comprised one trajectory. Using the results of this segmentation, we computed the average time and standard deviation of the time that elapsed between successive observations. Then, when the displacement time between two consecutive sensors was longer than the average time plus two standard deviations, we applied another segmentation in the daily trajectories. Finally, we only maintained trajectories bigger than two records in the dataset.

We assumed that drivers generally prefer the shortest path between two locations [15]. Therefore, we completed the transitions between two consecutive sensors according to the shortest path. In other words, if sensor $s_{j}$ appears after sensor $s_{i}$ in a trajectory, we assumed that all sensors on the shortest path from $s_{i}$ to $s_{j}$ were missing sensors; thus, we completed the trajectory considering these sensors.

Like in [11,16], we kept only the representative locations and discarded the sensors that occurred at less than the 25th percentile of frequency (18 observations) which may not be significant for global patterns. After applying this filter, the total number of sensors was 369. Figure 4 shows a histogram of the number of observations for each sensor in the final data, which depicts the unbalance between sensors’ frequencies. Finally, we evaluated the models using the holdout 80–20 approach, where we randomly chose 80% of trajectories for training and 20% for the test.

### 4.2. Embedding Models

Here, we briefly explain the NLP methods that were applied to generate the embeddings. In this work, the pre-trained embeddings are used to represent sensors in the location prediction task.

Word2Vec is a representation learning framework for words based on feedforward neural networks. Word2Vec presents two main architectures: the Continuous Bag-of-Words (CBOW) Model and the Continuous Skip-gram Model. In both architectures, word vectors are trained based on a slide window of n-grams on the corpus, such as $\left(w_{i-t},w_{i-t-1},\cdots w_{i}\cdots w_{i+t},w_{i+t+1}\right)$. The CBOW predicts the current word $w_{i}$ based on the window context $\left(w_{i-t},w_{i-t-1},w_{i+t},w_{i+t+1}\right)$. The Skip-gram predicts surrounding words $\left(w_{i-t},w_{i-t-1},w_{i+t},w_{i+t+1}\right)$ given the current word $w_{i}$. The main drawback of Word2Vec is that it is trained on a separate local context window instead of a global context.

The transformers architecture [35] introduced the self-attention mechanism, a sequence model that is able to create an embedding space of words that provides token representations based on the representation of the more relevant tokens in the sequence. BERT [28] is a multi-layer transformer encoder. that was proposed to mitigate the unidirectionality constraint of its previous language models. Unidirectionality implies that the model can only access the previous tokens in the self-attention Transformer layers. Contrary to unidirectionality, BERT creates an embedding space of words, considering both the left and right contexts.

The pre-trained word representations of BERT can be obtained using two different tasks, a masked language model (MLM), or a next sentence prediction. The MLM model randomly masks a percentage of the input tokens in the sequence. During training, the goal is to predict the masked tokens on the sequence based on their left and right contexts. This step results in the pre-trained representations for the tokens in a specific vocabulary. Word2Vec and BERT MLM are the solutions that were investigated in our experiments to learning sensor representations.

### 4.3. EST Prediction Model

Recurrent Neural Networks (RNN) [36] compose a class of neural networks that work on sequences of arbitrary lengths. Sequential data, such as trajectories, has a particular characteristic, where the order in which the instances appear in the sequence matters. Unlike Multilayer Perceptron (MLP), neural networks. which process each data input independently of the previous piece, RNNs can remember previous information in the sequence and update the network weights considering this past information. A Long Short-Term Memory Network (LSTM) [37] is an RNN tat is designed to overcome the vanishing and exploding gradient problem in vanilla RNN. An LSTM comprises several memory cells, with each one defining a hidden layer. A memory cell has a recurrent edge with a value associated with it, called a cell state.

The EST prediction model investigated, which was also based on the previous papers [14,15,16] is a recurrent based model. The first layer (1) is an Embedding Layer, which applies a linear transformation on the high-dimensional input vectors to reduce their dimensionality while trying to preserve the similarity between instances from the original space of features in the new feature space. The original representation of the input in our problem is the one-hot encoding representation, in which the dimensionality of the vectors is the number of existent sensors. A one-hot encoding represents categorical values as binary vectors, where all values are zero except the index of the categorical input, which is marked as one. The parameters of the embedding layer can be trained (or tuned) while training the other parameters in the neural network for the following location prediction or can be pre-trained using a representation learning model. The output of the embedding layer is sent to (2) a LSTM Layer, which is able to learn the moving patterns from the sensor sequence. In sequence, we have (3) a Dropout Layer, to help to prevent overfitting problems by randomly hiding some input units at each iteration of the training phase. Moreover, (4) an Output Layer is a fully connected layer. The activation function of the output layer is the Softmax. The Softmax function converts the output into a vector that indicates the probability of each sensor being the target. Figure 5 shows the EST prediction architecture that was utilized in our experiments.

Our EST prediction model received fixed-length sub-trajectories corresponding to the last *w* sensors that were observed. Thus, we used the sliding window strategy in the training step to convert the trajectories into fixed-length sensor sequences. In the sliding window (Figure 6), we scanned the entire trajectory by moving a window one position forward at each iteration. We began the window with the first sensor observation and retrieved the w=5 sensors in the window as features ($o_{1},o_{2},o_{3},o_{4},o_{5}$) and the first sensor outside the window, $o_{6}$, as the target. Iteratively, we moved the window and repeated the process until we found the last position in the trajectory.

## 5. Experimental Evaluation

From here, we conducted our research by splitting it into three sets of experiments: (i) Analysis of Location Prediction—investigate whether NLP models have the potential to model the vectorial space of features for location prediction; (ii) Analysis of Sensor Embeddings—investigate whether sensor embeddings obtained from NLP models can capture the relationship between sensor locations; (iii) Analysis of Trajectory Embeddings—investigate whether the trajectory embeddings obtained from NLP models can capture the behavior of the whole trajectory.

### 5.1. Analysis of Location Prediction

In this section, we conducted experiments to examine the research question RQ1: Could NLP embedding models, more specifically, language models and word embeddings, be used to represent the vector space of trajectories in location prediction tasks using recurrent architectures?

To extract the embedding representation of sensors, we evaluated both BERT MLM and Word2Vec. The embedding of words from NLP models was used as the embedding layer on top of the LSTM-based model (as shown in Figure 5) to predict the next location. The LSTM-based model was evaluated with and without fine-tuning on the embedding layer. The fine-tuning adjusts the weights of the embedding layer on the training step for one specific task (location prediction in our case). We also evaluated the pure LSTM model as a baseline, i.e., LSTM with an embedding layer that was trained from zero.

In BERT MLM, part of the trajectory sensors is masked, and the model is trained to predict the masked sensors considering the unmasked ones.

We propose applying a simple data augmentation strategy that works with a masked model. We replicated the training dataset a number of n_replication times. As in [28], we randomly chose the masked 15% of sensors on each replicated sample. In the case of Word2Vec, we applied CBOW architecture. The variation in hyper-parameters of BERT MLM and Word2Vec are shown in Table 1.

For the Word2Vec model, we varied the parameters (Table 1) and trained one EST prediction model using the embeddings obtained by each parameter configuration of Word2Vec. We reported the result for the configuration that acheived the best accuracy in the EST prediction model. For BERT MLM, we masked the last sensor in the validation data and collected the accuracy for each BERT MLM configuration. We trained an LSTM using the configuration that reported the best BERT MLM accuracy.

Using the pre-trained embeddings, we trained LSTM models to predict the next sensor given the previous *m* (we varied *m* according to the set [5, 7, 15, 31]; when *m* = 5, the model outperforms others) observed sensors on the same trajectory. The models were evaluated using ACC@N and closeness_error metrics. The ACC@N measures the percentage of instances where the correct prediction was among the *N* most probable outputs according to the estimation given by the model, using the softmax activation. The closeness_error is the road network distance between the predicted sensor and the expected one.

**Models**. Concerning the models in our work, we considered the following configurations: (i) LSTM, where the embedding layer was trained from scratch on the prediction task; (ii) LBERT, where the LSTM usined the pre-trained embeddings of BERT MLM as an embedding layer; (iii) LBERT-FT, where the pre-trained BERT embedding layer was fine-tuned under the next sensor prediction task; (iv) LW2V, where the LSTM used the pre-trained embeddings of Word2Vec as an embedding layer; (v) LW2V-FT, where the pre-trained Word2Vec embedding layer was fine-tuned under the next sensor prediction task. Furthermore, we only report the results of the better parameter configurations.

Table 2 presents the results for the best model configuration for each strategy. The best accuracy for each different window size was reported in Appendix A. The LSTM predictor with the fine-tuned BERT embedding layer achieves the best result. The fine-tuning of the BERT-embedding layer improved the accuracy ACC@1 by up to 7%; the same behavior was found for ACC@2 and ACC@3. We believe that BERT better represents the feature space of sensors for trajectories because it learns representations by adjusting the weight of the context. In addition, the positional encoder of BERT captures sequential connexions from the road network. The masked language model of BERT also simulates and learns under the situation of missing data, where the sensor does not capture the passage of a moving object. The LSTM predictor using Word2Vec embedding reached the lowest accuracy, 52.14%, followed by its fine-tuned version. The fine-tuning of the Word2Vec embedding layer did not significantly improve the predictor. One possible reason is that Word2Vec models do not learn sequential patterns, but only the surrounding context. The baseline LSTM achieves 66% of accuracy, these results are consistent with the ones obtained by [16].

Table 3 reports the mean and percentiles of the closeness error obtained by the models. The median and lower percentiles were omitted since, for all models, their values were zero. The mean closeness error of LBERT-FT was the best one, with a value of 0.79 km, followed by LBERT with 0.97 km and LSTM with 1 km. LW2V and LW2V-FT found the worst results, in this order. LBERT-FT obtained 80% of all predictions with errors under 1.3 km. LBERT reached a 70% lower of closeness error than 0.5 km and an 80% lower closeness than 1.6 km. We can conclude that BERT representation not only helped to increase the accuracy, but also increased the proximity between predicted and expected sensors when the predictor failed. However, with fine-tuning for location prediction tasks, BERT representations could achieve better results.

### 5.2. Analysis of Sensor Embeddings

These experiments are guided by the research question RQ2: Are the representations of sensors/locations from representation models in NLP that can capture their context, i.e., the closest sensors/locations, in terms of both road distance and connectivity?

Our goal is to evaluate if the spatial relationship impacts the degree of similarity of the sensor embeddings. We evaluate the relationship between spatial and embedding distances and investigate how similar embeddings reflect the spatial proximity or connectivity in the road network. To achieve this goal, we used BERT MLM, which leverages the best next-location predictor from the experiments, as explained in the previous section (Section 5.1). BERT MLM was used to generate the embedding space of the sensors. Figure 7 exemplifies the embedding vectors of sensors in a space. The points (in red) are sensors, and the markers (in black) connected by a line represent a pair of nearest sensors according to the cosine distance between their embedding vectors.

Although the nearest embedding sensors vectors from BERT MLM are not directly the closest in space, we can see, in Figure 7, that they are seen on the roads with some connectivity and also in their spatial neighborhood. This suggests that BERT MLM could capture some spatial relationship between sensors.

To better understand the impact of spatial proximity on embedding similarity, we analyze the Mean Reciprocal Rank (MRR) of sensors concerning these metrics. In Information Retrieval, the Reciprocal Rank (RR) calculates the rank at which the first relevant document was retrieved. If the relevant document was retrieved at rank *r*, then RR is 1/r [38]. We adapt the definition of RR measure for our particular goal (Definitions 5 and 6).

**Definition** **5**(Reciprocal Rank). *Given a reference distance $d_{ref}$, a query distance $d_{query}$, a query object o and a set of objects O, the reciprocal rank of o with respect to the reference distance $d_{ref}$ and the query distance $d_{query}$ for the set O is 1/r, if the closest object to o, $o^{*}\in O$ according by $d_{ref}$ is at rank r according by $d_{query}$.*

In these experiments, the reference distance $d_{ref}$ was set to be a spatial distance (Euclidean and road distances), while the query distance $d_{query}$ was the cosine distance between embedding vectors.

**Definition** **6**(Mean Reciprocal Rank). *The Mean Reciprocal Rank (MRR) is the average RR across a set of query objects.*

First, we collected the MRR using the Euclidean distance as $d_{ref}$, and Cosine distance as $d_{query}$. In that case, the MRR was 0.20, which means the nearest sensor according to Euclidean distance is around the 5th most similar on the embedding space, on average. Using the Road Network distance as $d_{ref}$ and cosine distance as $d_{query}$, MRR was 0.17, so the nearest sensor in the road network is, on average, around the 6th most similar on the embedding space. We believe that, on average, being in the top five or six is acceptable, as we are learning a new space (embeddings) that can represent the sensors, which is not a trivial task.

We also evaluated the variation in MRR for sensors inside a neighborhood using the following methodology. We calculate the RR for each sensor considering a spatial distance (both the Euclidean and the Road Network distances) as the reference distance and the cosine distance between the embedding vectors as the query distance. The set of sensors to be ranked (*O*) were filtered, leaving only the ones that were spatially close to the sensor considered as the query object, i.e., the ones that fell inside the distance range. We varied the range distance and collected the MRR according to the filtered sensors.

Figure 8 presents the result of this analysis for both Euclidean and road network distances. One can observe that, among pairs of sensors with a distance of around 1.5 km, the MRR is about 0.35, which means that the closest sensor in space was found at rank 2 or 3; when the filter of maximal distance increases, the MRR decreases. Even when the maximal distance is around 4 km for Euclidean space, the MRR is more significant than 0.25, and for the road distance of approximately 4 km, the MRR is 0.20. In other words, the rank given by the embeddings of the closest sensor is around 4 and 5 for euclidean and road distances, respectively. The embedding representation reached a rank that was slightly similar to the spatial rank for the set of sensors that are not as spatially distant. We argue that an efficient model of sensor representation does not necessarily reflect only the spatial proximity, but may also reflect the connectivity in terms of road connectivity and frequent paths. Overall, we note from the previous research question that BERT sensor embeddings achieved better results for the next location prediction task. Furthermore, the BERT embeddings tend to reflect more spatial similarity when considering the sensor’s neighborhood as limited to a distance.

### 5.3. Analysis of Trajectory Embeddings

In this section, we investigated the research question RQ3: Could trajectory representations from NLP embedding models adequately capture trajectories’ similarity? This was conducted by evaluating the similarity between trajectory embeddings regarding the similarity between raw trajectories.

We defined trajectory embedding as the average vector of the embedding of the sensors comprising the trajectory. As in Section 5.2, we used BERT MLM to generate the embedding representation of sensors. Figure 9 exemplifies different pairs of the most similar trajectories in the test set according to the cosine distance of their embedding vectors. Consider the trajectory source, the marker with a triangle, and the target, the other marker. We divided the figure into three cases: when trajectories present high, medium, and low spatial similarity.

We aim to show that the trajectory embeddings can capture the spatial likeness for some cases but not for others. A further direction to improve the embedding quality is training with more data or investigating other language models.

We evaluated the Reciprocal Rank and the Mean Reciprocal Rank (Definitions 5 and 6) between raw EST trajectories and their embeddings using the Dynamic Time Warping (DTW) and the Edit (ED) distances between raw trajectories, and also the cosine distance between embedding vectors. DTW is one of the most popular trajectory distance measures. It searches for all possible alignment points between two trajectories to find the one with minimal distance. To measure the distance between sensors, we used the Euclidean distance. ED quantifies the dissimilarities between two sequences of strings by counting the number of operations that are needed to transform one string into another. For more details about trajectory similarity functions, we refer to the survey [39].

The experiment considers the cosine distance between trajectory embeddings as the query distance. We perform two experiments: compute the MRR using DTW as a reference distance and another one using ED as a reference distance. By using DTW as the reference distance, MRR obtained the best results, with a value of 0.44. In other words, the embedding rank for the most similar trajectory according to DTW is between 1 and 3, on average. The MRR using ED as the reference distance was 0.27. We believe that being on the top from one to three, on average, is a good result, as we are learning a new embedding space to represent trajectories, which is a complex task due to the nature of these trajectory data.

Another experiment we performed was filtering out the set of trajectories objects *O* to contain only those with the reference distance under a maximal value. Similar to the analysis of sensor embeddings, we evaluated how the MRR performs on a subset of trajectories when the neighborhood increases. Figure 10 shows the results. For the most similar trajectories pairs (DTW distance of approximately 5), the MRR was close to 5. For trajectories on the neighborhood filtered by DTW equal to 85, the MRR reached the minimum value (0.44) and remained almost constant when the neighborhood increases. This result implies that BERT MLM embeddings could capture spatial similarity and connectivity when the sensors of similar trajectories did not exactly match. The highest value of MRR using ED was 0.33, at the point at which ED was at a maximum of 10. This result implies that BERT embedding can represent the sequence of discrete location labels in a trajectory (measured by edit distance). With these experiments, we achieved our goal of using an NLP model and evaluating its quality to represent the trajectories and capture their spatial similarity.

## 6. Conclusions and Future Works

In this paper, we investigated the use of NLP models to generate the embedding space of features for external sensors’ trajectories. We analyzed the quality of the embedding trajectories using extrinsic and intrinsic strategies. The extrinsic approach is concerned with evaluating a trajectory prediction architecture using different embedding vectors. In the intrinsic evaluation, we analyzed the reciprocal rank of embedding vectors using well-known distance measures for locations (in our case, sensors) and trajectories. The experimental evaluation has shown the applicability of BERT MLM to extract embedding vectors and represent trajectories. BERT MLM demonstrated the best result for location prediction tasks, mainly when fine-tuning the embedding layer. The reciprocal rank analysis showed that embeddings can represent the spatial similarity for locations in a restricted region surrounding the area. However, when the neighborhood expands, the embedding distance among sensors does not maintain its correlation with spatial distance (see Figure 8).

Additionally, when the query range of trajectories increases, the correlation between trajectory similarities and the embedding distance does not considerably decrease (see Figure 10). In future work, we propose to investigate distinct approaches to compare and evaluate trajectories’ embeddings under different tasks and metrics. In future work, we also aim to investigate how to apply few-shot learning approaches to NLP models in the context of external sensor trajectories.

## Figures and Tables

**Figure 1 sensors-22-07475-f001:**
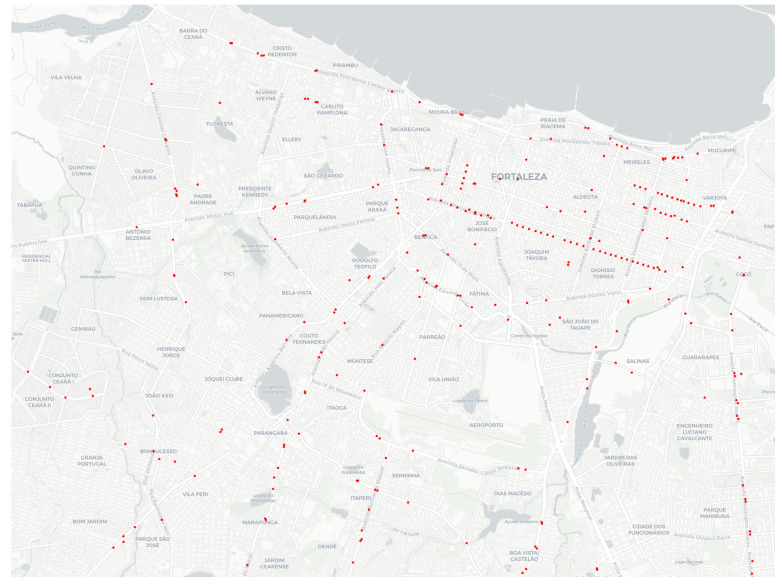
Map of Fortaleza city with external sensors (surveillance cameras) positioned on the road network in 2019.

**Figure 2 sensors-22-07475-f002:**
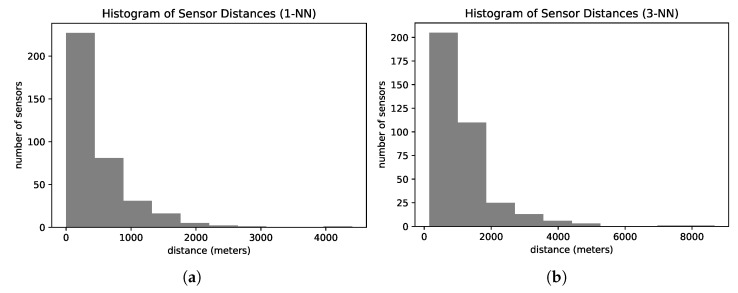
Histograms of road distances to nearest sensors on the road network. (**a**) Distances to the 1st nearest sensor. (**b**) Distances to the 3rd nearest sensor.

**Figure 3 sensors-22-07475-f003:**
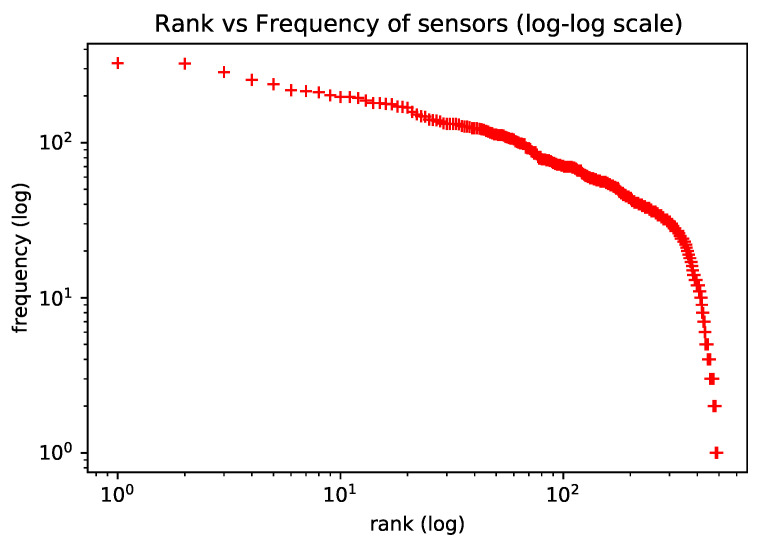
Statistics of sensor observations satisfy Zipf’s Law.

**Figure 4 sensors-22-07475-f004:**
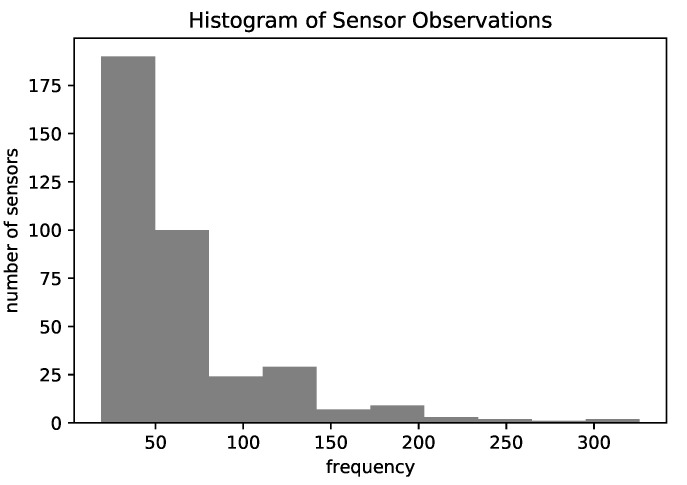
Histogram of the number of observations by each sensor.

**Figure 5 sensors-22-07475-f005:**
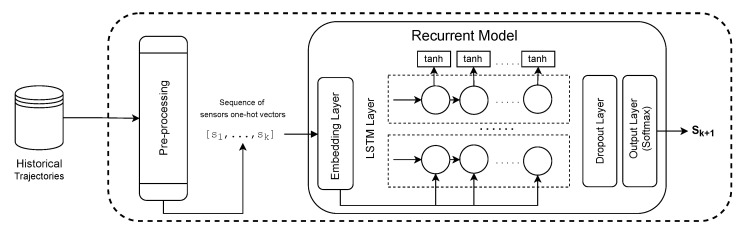
LSTM EST prediction architecture.

**Figure 6 sensors-22-07475-f006:**
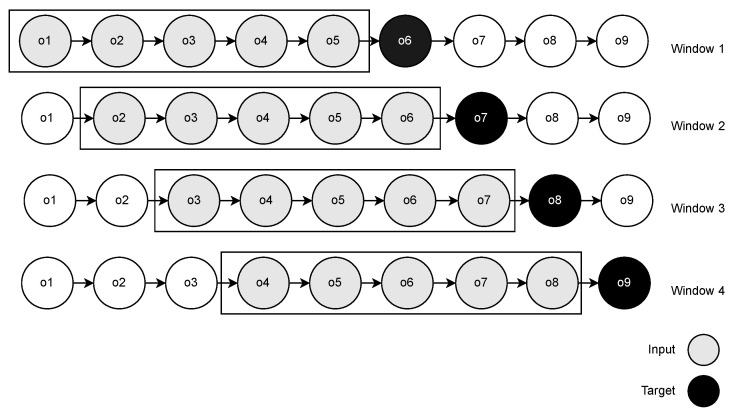
Sliding window approach.

**Figure 7 sensors-22-07475-f007:**
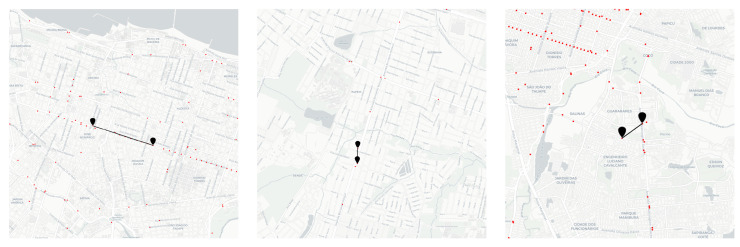
Example of nearest sensors according cosine distance between BERT MLM embedding vectors.

**Figure 8 sensors-22-07475-f008:**
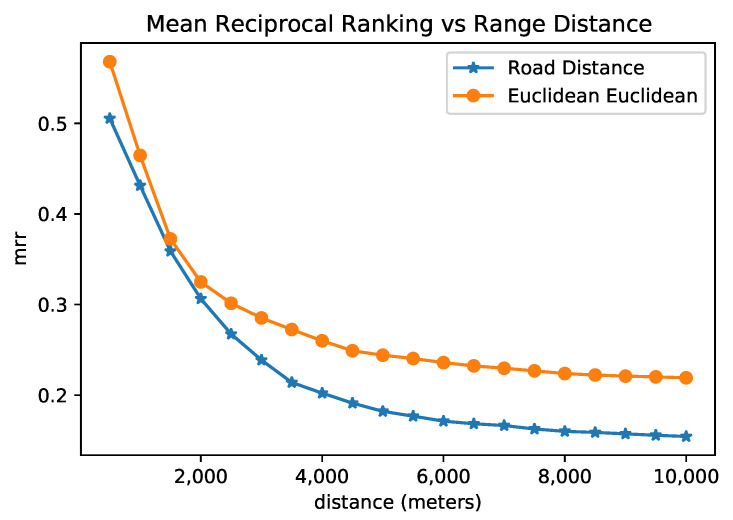
Mean reciprocal rank of embeddings among sensors inside a neighborhood in space.

**Figure 9 sensors-22-07475-f009:**
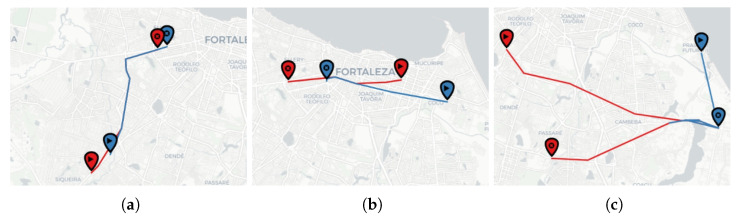
Example of most similar trajectories according to cosine distance between their embedding vectors. (**a**) High spatial similarity. (**b**) Medium spatial similarity. (**c**) Low spatial similarity.

**Figure 10 sensors-22-07475-f010:**
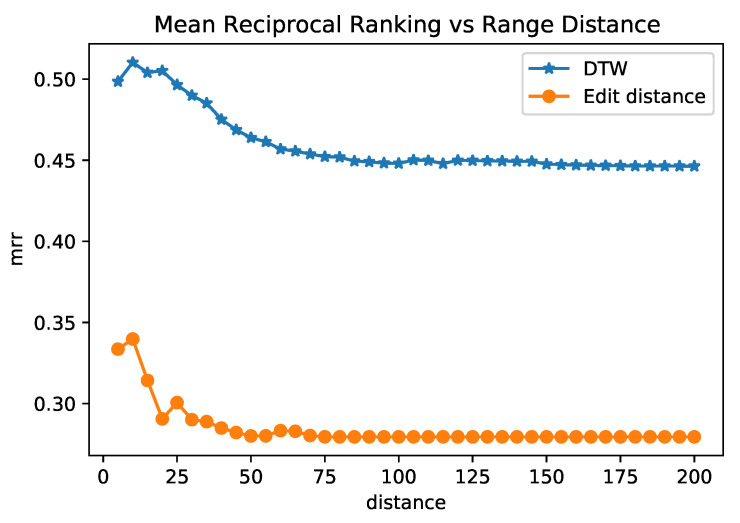
Mean reciprocal rank of the embeddings among trajectories pairs under a maximal distance.

**Table 1 sensors-22-07475-t001:** Word2Vec and BERT MLM Parameters.

Word2Vec	BERT MLM
embedding size = [16, 32, 64, 128]	embedding size = [16, 32, 64, 128]
n-grams = [4, 6, 8, 10]	sequence size = [32, 64]
	intermediate sizes = [64]
	hidden dims = [64, 128, 256]
	num hidden layers, num attention heads = [8]

**Table 2 sensors-22-07475-t002:** Accuracy of prediction models.

	ACC@1	ACC@2	ACC@3
LSTM	64.10	74.96	80.32
LBERT	66.84	79.70	85.05
LBERT-FT	73.71	85.35	90.01
LW2V	52.14	63.09	72.38
LW2V-FT	53.33	66.90	72.14

**Table 3 sensors-22-07475-t003:** Mean and percentiles of the closeness error for predictions (in km).

	Mean	60	70	80	90
LSTM	1.0	0.0	0.85	1.99	3.49
LBERT	0.97	0.0	0.56	1.67	2.78
LBERT-FT	0.79	0.0	0.0	1.36	2.78
LW2V	1.44	0.73	1.4	2.44	4.09
LW2V-FT	1.36	0.65	1.31	2.46	4.09

## Data Availability

Due to confidentiality agreements, supporting data can only be made available to bona fide researchers subject to a non-disclosure agreement.

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
