# Peer review of "Modeling Trajectories Obtained from External Sensors for Location Prediction via NLP Approaches"

_sensors, 2022, doi:10.3390/s22197475_

Round 1

Reviewer 1 Report

The manuscript proposes to leverage the representation learning methods in NLP to learn the features of external sensors as well as the trajectories, and further examine the performance of the method in location prediction task. In general, the problem to be addressed is very interesting and also important. However, there are still some issues need to be addressed for improvement. The specific comments are listed as follows:

1.     The title of the manuscript seems to be too vague and general. I would like to suggest narrowing down the scope with more details.

2.     In the introduction section, the background introduction and literature review are not well organized, which should be more carefully re-structured.

3.     In Section 2, the authors define the concept of “EST Prediction”, which basically represents the problem of road network-constrained vehicle next location predication with the spatial resolution of sensors’ locations. So, what is the connections and differences between this task and general next location prediction tasks as reviewed in Section 3.2? What is special about the task per se?

4.     In Section 3.1, the title may be replaced with “Representation Learning in NLP” rather than the very general “Representation Learning” to narrow down the scope. In addition, the first sentence indicates that “Representation learning refers to the unsupervised machine learning methods that …”, which however is not very accurate. The term “representation learning” is quite broad and may not just refer to unsupervised methods, and it also includes supervised and semi-supervised learning methods with the goal of learning representations/features of specific targets.

5.     In Section 3.1, TraceBERT is introduced, and the authors claim that “TraceBERT follows the same idea as ours”. However, the authors do not highlight their differences or contributions compared with the previous study. It is better to explicitly compare with it.

6.     In Section 4.1, it is better to visualize the data set to give us an overview of the complexity of the data and also the task. For example, how complex is the road networks as well as the associated sensors.

7.     In Section 4.1, the setup of the data set for training and evaluation is not elaborated. More details are needed.

8.     For the data preprocessing in Section 4.1, the authors write that “Finally, we keep only the sensors that occur more frequently than average in the training set”. What is the rationale and motivation behind the preprocessing procedure? Is there any reference or justification for it? Will the step reduce the difficulty in location prediction?

9.     In Section 4.2.1, the authors state that “In the training step, the trajectories are converted into fixed-length sequences of sensors”. Is this means that each trajectory will be processed into the same length? If so, why is this process necessary and how to implement this process?

Author Response

Dear Reviewer,

Thank you for your time and effort to provide feedback on our paper. Please find a detailed discussion of your concerns below.

1- The title of the manuscript seems to be too vague and general. I would like to suggest narrowing down the scope with more details.

2- In the introduction section, the background introduction and literature review are not well organized, which should be more carefully re-structured.

 Response to comments 1 and 2. Thanks for your suggestion. We changed the title of the paper and arranged the introduction according to the following topics: location prediction problem (which is the problem we tackle in the paper), representation learning, natural language approaches for representation learning, and some related works to our problem. 

 3- In Section 2, the authors define the concept of “EST Prediction”, which basically represents the problem of road network-constrained vehicle next location prediction with the spatial resolution of sensors’ locations. So, what is the connections and differences between this task and general next location prediction tasks as reviewed in Section 3.2? What is special about the task per se?

 Response to comment 3. Thank you for your question. Location prediction from external sensors has to deal with trajectory data much more sparse than the usual GPS raw trajectories. Also, instead of continuous locations, we have discrete ones, as in next-stop prediction using check-ins or identified stops. Usually, for next-stop prediction,  the application has available some additional information, like user profile, user id, semantic tags of the POIs, or text from the check-in application, which are not available in our work. Furthermore, unlike next-stop prediction, the permanency time at locations is too short since the objects are moving on the streets. In this kind of data, the sensors can fail to capture the vehicle plates, generating incomplete trajectories, thus we have a missing data problem. The objects move in the road network as in the path prediction problem, but different from the path prediction problem, we do not have the track of the complete path available. We improved this discussion in the new version of the paper. 

 4- In Section 3.1, the title may be replaced with “Representation Learning in NLP” rather than the very general “Representation Learning” to narrow down the scope. In addition, the first sentence indicates that “Representation learning refers to the unsupervised machine learning methods that …”, which however is not very accurate. The term “representation learning” is quite broad and may not just refer to unsupervised methods, and it also includes supervised and semi-supervised learning methods with the goal of learning representations/features of specific targets.

 Response to comment 4. Thank you for your suggestions. The related work in Section 3.1 is about representation learning for mobility data, using or not NLP models. We applied your suggestion to narrow down the scope, and change the name to “Representation Learning for Mobility Data”. 

 5- In Section 3.1, TraceBERT is introduced, and the authors claim that “TraceBERT follows the same idea as ours”. However, the authors do not highlight their differences or contributions compared with the previous study. It is better to explicitly compare with it.

 Response to comment 5. Thank you for your suggestion. TraceBERT applies BERT Masked Language Model to complete trajectories for the trajectory reconstruction problem, which aims to predict missing locations along the trajectory. The original task of BERT MLM is to predict masked tokens on a text, which is used to learn the pre-trained embedding representation. In TraceBERT, each location is modeled as a token and the tokens are masked. Thus, in TraceBERT they evaluate if BERT MLM is able to complete the missing locations, which is basically the same task of BERT applied to the trajectory domain. In our work, we use BERT MLM and Word2Vec to pre-trained embeddings of sensors. We use the obtained embeddings in the EST prediction task, by incorporating the pre-trained embeddings into an LSTM model. We evaluate the performance of prediction models using the pre-trained embeddings and using a fine-tuning strategy. We also evaluate the best embedding of locations and trajectories via intrinsic evaluation, which uses metrics on the embeddings obtained itself.  We detailed these differences in the new version of the paper. 

 6- In Section 4.1, it is better to visualize the data set to give us an overview of the complexity of the data and also the task. For example, how complex is the road networks as well as the associated sensors.

 Response to comment 6. Thank you for your suggestion. We plotted a map of sensors on the road network and some statistics about the data set in the new version of the paper. 

7-  In Section 4.1, the setup of the data set for training and evaluation is not elaborated. More details are needed.

Response to comment 7. Thank you for this comment. We have used the holdout approach 80/20, which randomly chooses 80% of trajectories for training and the last 20% for evaluation. To train the LSTM model, we also applied the sliding window approach. We included this in the new version of the paper. 

 8-  For the data preprocessing in Section 4.1, the authors write that “Finally, we keep only the sensors that occur more frequently than average in the training set”. What is the rationale and motivation behind the preprocessing procedure? Is there any reference or justification for it? Will the step reduce the difficulty in location prediction?

Response to comment 8. Thank you for your question. The idea is to maintain only the most representative data and discard outliers. There are several ways to apply this idea to train the machine learning model for location prediction. One can keep only more representative locations, like in our case, or the more representative users (when the user is available for the model). One can also keep only the representative trajectories, by filtering by size or time duration. TraceBERT[11] and  [16]  do a similar pre-processing by discarding rare locations. SERM[19] discards users that have less than 50 records. [20] discard non-active users, considering the active users with more than 50 check-ins. In our case, we discard the sensors that occur less than the 25th percentile of frequency (18 observations). We correct this information and discussed it a little more in the new version of the paper.

9 - In Section 4.2.1, the authors state that “In the training step, the trajectories are converted into fixed-length sequences of sensors”. Is this means that each trajectory will be processed into the same length? If so, why is this process necessary and how to implement this process?

 Response to comment 9. Thank you for your comment. Our LSTM model receives trajectories in a fixed-length, concerning the sub-trajectory of the last w observed sensors before the target sensor (the one that has to be predicted). So, we used the slide-window strategy, very known in time-series and sequence processing. In the sliding window approach, we scan the whole trajectory moving a window of sensors. We start the window in the first sensor and retrieve the w sensors in the window to compose the sub-trajectory. Then, we move the window to the second sensor and retrieve w sensors again. Iteratively, we move the window to the next sensor and do the same in each iteration until finding the last position in the trajectory. We discussed more and provided an illustration to clarify this process in the new version of the paper.  

Kind regards.

Lívia A. Cruz

Reviewer 2 Report

This paper studies a sensor track prediction problem in block road network. In the block road network, because each sensor can only observe the moving target within a certain range, this paper predicts the next sensor range that the target may enter according to the historical observation record of a target and the position data of all sensors. In this paper, all sensors are numbered, and each sensor is likened to a word, and then the word hopping model in the word vector model of NLP method is used for modeling. The model structure is implemented by RNN network, and the model is trained by the observed data. The training results are given in this paper.

There are still several problems with the paper:

1. Training data: The general NLP method requires a large number of words and sufficient training samples. However, the training sample size in the paper is small, please explain how to get a better training model with small data sample;

2. Model prediction accuracy: The Word2Vec Model proposed in this paper is an open-loop model. How to ensure the accuracy of sensor prediction? The precision definition and corresponding criterion of sensor prediction are not given in the paper. 

So in my opinion, I suggest that this paper in this version cannot be published, which need a major revision.

Author Response

Dear Reviewers,

Thank you for your time and effort to provide feedback on our paper. Please find a detailed discussion of your concerns below.

1- Training data: The general NLP method requires a large number of words and sufficient training samples. However, the training sample size in the paper is small, please explain how to get a better training model with a small data sample.

Response to comment 1. Thank you for your comment. In some NLP papers, there are low-resource languages with small annotated datasets compared to English and Chinese, in which some deep learning techniques and few-shot learning   are applied, like in these papers:

Gu, J., Wang, Y., Chen, Y., Cho, K., & Li, V. O. (2020, January). Meta-learning for low-resource neural machine translation. In 2018 Conference on Empirical Methods in Natural Language Processing, EMNLP 2018 (pp. 3622-3631). Association for Computational Linguistics.

Brown, T., Mann, B., Ryder, N., Subbiah, M., Kaplan, J. D., Dhariwal, P., ... & Amodei, D. (2020). Language models are few-shot learners. Advances in neural information processing systems, 33, 1877-1901. 

In future work, we can investigate how to apply few-shot learning approaches in the context of external sensor trajectories. We also highlight that we changed the configuration of the BERT MLM model to reduce the number of parameters, which is more suitable for small data. The parameters in BERT MLM architecture are available in Table 1 in the paper. Moreover, as we know to train the BERT MLM model, part of the training trajectory sensors should be masked, and the model is trained to predict the masked sensors considering the not-masked ones. In the paper, we proposed applying a simple data augmentation strategy that works with a masked model. We replicated the training dataset a number n of times. As in [28], we randomly chose to mask 15% of sensors on each replicated sample. 

2- Model prediction accuracy: The Word2Vec Model proposed in this paper is an open-loop model. How to ensure the accuracy of sensor prediction? The precision definition and corresponding criterion of sensor prediction are not given in the paper. 

Response to comment 2. Thank you for your question. To generate the embedding space of sensors using the Word2Vec approach, we varied Word2Vec parameters (available in Table 1). Then, we evaluate the LSTM predictor using many variations of Word2Vec, like in a grid search. Then, we chose the result that achieves better final accuracy. So, we estimated the quality of Word2Vec embeddings using our specific task, not the accuracy of Word2Vec itself. However, it is also possible to get the accuracy of the Word2Vec approach. For this, we refer to the method available in the Gensim library that provides an implementation of this  https://tedboy.github.io/nlps/generated/generated/gensim.models.Word2Vec.accuracy.html.  We explain this better in the new version of the paper. 

Concerning the definition of sensor prediction, please refer to Definition 4, which was improved in the new version of the paper. 

Kind regards.

Lívia A. Cruz

Round 2

Reviewer 1 Report

The revision has well addressed my previous concerns. The authors may further check and revise some minor spelling issues at their own discretion.

Author Response

Dear reviewer, 

Thank you for your availability to review our article. 
We uploaded a new version with some improvements in the style and spelling of English. All the recent changes are in blue, while the previous updates are still in red.

Best regards.
On behalf of the authors,
Lívia A. Cruz

Reviewer 2 Report

In this revised version, the experimental data is full and the analysis of the experimental results is very profound, which can reasonably support the conclusion.

Therefore, it is recommended to accept the paper.

Author Response

(The authors gave the same response as above.)
